# Effect of Lactic Acid Bacteria on the Nutritive Value and In Vitro Ruminal Digestibility of Maize and Rice Straw Silage

**Tabita Dameria Marbun** [1], **Kihwan Lee** [1], **Jaeyong Song** [1], **Chan Ho Kwon** [1], **Duhak Yoon** [1], **Sang Moo Lee** [1], **Jungsun Kang** [2], **Chanho Lee** [2], **Sangbuem Cho** [3] **and Eun Joong Kim** [1,*]

[1] Department of Animal Science, Kyungpook National University, Sangju 37224, Korea; tabitamarbun@gmail.com (T.D.M.); zoosuk87@gmail.com (K.L.); jysong1976@gmail.com (J.S.); chkwon@knu.ac.kr (C.H.K.); dhyoon@knu.ac.kr (D.Y.); smlee0103@knu.ac.kr (S.M.L.)

[2] Genebiotech Co., Ltd., 166 Sinwonsa-ro, Gyeryong-myeon, Gongju 32619, Korea; jskang@genebiotech.co.kr (J.K.); chlee@genebiotech.co.kr (C.L.)

[3] Circulation Agriculture Livestock Solution Inc., Sangju 37224, Korea; chosb73@gmail.com

\* Correspondence: ejkim2011@knu.ac.kr; Tel.: +82-54-530-1228

**Abstract:** A study was conducted to determine the effects of lactic acid bacteria (LAB) on nutritive value and in vitro rumen digestibility of maize and rice straw silages. Two identical experiments were carried out for each of the two silages. A total of five treatments were used for each experiment: (1) negative control (NC); (2) positive control (PC); (3) *Lactobacillus plantarum* (LPL); (4) *L. paracasei* (LPA); and (5) *L. acidophilus* (LA). Each treatment was then divided into four ensiling periods: 3, 7, 20, and 40 days with three replications. The LPL treatment had significantly higher dry matter (DM), lower ammonia-N, and a lower number of fungi on maize silage after 40 days ($p < 0.05$). On the other hand, the LA treatment increased DM and CP content, reduced NDF and ADF contents compared to NC, and also produced more lactic acid compared to the other LAB-treated rice straw silages. Results of the in vitro rumen fermentation of maize silages showed no significant differences in DMD after LAB inoculation. However, higher DMD and ruminal ammonia-N were shown by rice straw ensiled with *L. acidophilus*. In conclusion, silage additives, which could improve the ensiling process of maize and rice straw, appeared to be different and substrate specific.

**Keywords:** rice straw; maize; silage; lactic acid bacteria

## 1. Introduction

Forage is an important source of nutrients in the diets for ruminants worldwide, providing energy and protein for maintenance, growth, and production [1,2]. Unreliable weather conditions make it difficult for farmers to produce good quality forage year-round [3] and, as a result, various methods have been used to preserve fresh forages (e.g., hay or silage).

Ensiling is one of the methods used to preserve fresh forages. Silage has several benefits which include low and stable pH (from the production of acids during the ensiling process) as well as improved dry matter intake (DMI) [4–6]. To facilitate the fermentation process during ensiling, silage additives such as lactic acid bacteria (LAB) have been added [4,7]. Inoculation of silage substrate with LAB results in a rapid decline in pH at the beginning of the silage fermentation [8,9] with a concomitant increase in lactic acid production [9]. Furthermore, there is a reduction in ammonia-N formation in LAB inoculated substrate compared with uninoculated substrates [4]. There are several different strains of LAB available for use during silage making, each with its own unique fermentation

characteristics [10,11]. A number of silage additives are commercially developed with mixed strains of LAB, but they may be lacking in substrate specificity for certain roughages such as rice straw [12].

In the current study, two substrates with different nutritional characteristics were used to determine the effects of silage inoculants on their nutritive value. Maize (*Zea mays* L.), grown as a forage feed worldwide [13], is generally preserved as silage because of its low dry matter (DM) content (28–40%), low buffering capacity, and substantial water-soluble carbohydrates which produce organic acids, including lactic acid [11,14]. Feeding maize silage can increase the forage DM intake [15–17] and the milk yield as it is rich in metabolizable energy [16,17]. On the contrary, rice straw (*Oryza sativa*), an agricultural by-product, has relatively low digestibility due to a high level of cell wall components such as cellulose, hemicellulose, and lignin [18,19]. In Asian countries, ensiling rice straw was also one alternative to re-utilize this byproduct instead of disposing of it by burning and causing air pollution [20,21]. In the study by Pi et al. [22], feeding rice straw-based total mixed ration to Boer goats can produce favorable carcass characteristics and meat quality.

The fermentation characteristics of maize and rice straw differ due to their differing nutrient profiles and, as such, the application of appropriate additives for ensiling is essential to get the best out of each forage. The aim of the study was to examine how different LAB strains added during the ensiling process affect the nutritive value and in vitro digestibility of maize and rice straw silages.

## 2. Materials and Methods

### 2.1. Experimental Design and Ensiling Process

Two different silages were prepared using maize and rice straw. Experiment 1 was conducted in Gunwi-gun, Korea, in August 2013 using ripe, yellow maize (*Zea mays* L. cultivar Kwangpyeongok), while Experiment 2 was carried out in Cheongni-myeon, Sangju, Korea, in October 2013 using rice straw (*Oryza sativa* subsp. Japonica, cultivar Ilpum) as the substrate. Approximately 300 kg (fresh basis) of each substrate were harvested, chopped, and set aside for treatment.

Five treatments were applied to each of the maize and rice straw. Among the treatments there were two controls: a negative control, treatment of the forages with distilled water (NC) and a positive control, treatment of the forages with a commercial silage additive containing *L. acidophilus* ($4.4 \times 10^8$ CFU/g), *L. plantarum* ($9.5 \times 10^8$ CFU/g), and *Pediococcus acidilactici* ($8.9 \times 10^{10}$ CFU/g) (PC). The remaining three treatments were LAB treatments with the following species: *L. plantarum* (LPL), *L. paracasei* (LPA), and *L. acidophilus* (LA). All LAB treatments had $10^7$ CFU/g of LAB in 50 mL of distilled water. After administering the treatments, the maize and rice straw were put into 10 L plastic containers, sealed, and ensiled in triplication at room temperature for 3, 7, 20, and 40 days before opening.

### 2.2. Sampling Procedure

Samples of the silage were made using the coning and quartering method, and representative samples were placed in $35 \times 76.5$ cm net bags (1 mm$^2$ pore size) and oven dried at 65 °C for three days to determine initial dry matter (DM) content. After drying, the samples were weighed and ground small enough to pass a 1-mm screen using an ultra-centrifugal mill ZM 2000 (Retch, Haan, Germany) in preparation for proximate analysis. A 200 g aliquot of fresh samples was stored in sealed plastic bags at −20 °C for further analyses.

### 2.3. Chemical Analyses

For the determination of the chemical composition of both silages, DM (AOAC: 934.01), organic matter (OM; AOAC: 942.05), crude protein (CP; AOAC: 984.13), and ether extract (EE; AOAC: 920.39A) were determined using the AOAC procedures [23]. Neutral detergent fiber (NDF) and acid detergent fiber (ADF) analyses were conducted based on the methods of Van Soest et al. [24] using an ANKOM$^{2000}$ Fiber Analyzer (ANKOM Technology, Macedon, NY, USA). The frozen silage samples were thawed,

and approximately 100 g of silage was mixed with 900 mL of distilled water and homogenized using a stomacher (Daihan Scientific, Wonju, Korea) for about 4 min [14,25]. An aliquot of 90 mL of the silage extracts was taken and pH was immediately measured by using probe (8157BNUMD ROSS Ultra pH/ATC Probe, Thermo Orion Inc., Massachusetts, United States) connected to pH meter (Orion 3 Star, Thermo Scientific, Waltham, MA, USA). The extracts were then used for ammonia-N and organic acids analyses. The ammonia-N concentration was determined according to the method of Chaney and Marbach [26]. The high-pressure liquid chromatography method, according to Canale et al. [27], was used to determine organic acid profiles. A $300 \times 7.80$ mm Rezex ROA-Organic Acid $H^+$ (8%) column (Phenomex, Torrance, CA, USA) and a guard column (SecurityGuard, Cartridge Kit, Phenomenex, Torrance, CA, USA) were used with 0.005 N $H_2SO_4$ as the mobile phase with a 0.5 mL/min flow rate at room temperature.

*2.4. Microbial Analyses*

A sample of the silage (25 g) was chopped to a size of 2 cm and mixed with 225 mL saline solution (0.9% NaCl). This mixture was then homogenized for about 2 min. The homogenized silage–saline mixture was then serially diluted and spread on different types of media. For detection of Lactobacillus and Bacillus, Difco™ Lactobacillus MRS agar and Difco™ LB agar Miller (Becton, Dickinson and company, Sparks, MD, USA), respectively were used and samples were incubated for 24 h at 35 °C, while for yeast and fungi determination, Difco™ potato-dextrose agar (Becton, Dickinson and company, Sparks, MD, USA) was used and incubation was carried out for 36 h at 30 °C [28,29].

*2.5. In Vitro Rumen Fermentation Study*

In vitro rumen fermentation was conducted using the method by Tilley and Terry [30]. Half a gram of maize silage at 40 days of ensiling and rice straw silage at 40 days was weighed and put in incubation bottles. Rumen contents were obtained from two cannulated Hanwoo steers fed total mixed ration (TMR, dry matter 75.5%, crude protein 9.6%, crude fiber 15.82%, neutral detergent fiber 32.7% and acid detergent fiber 18.6%) and filtered using four layers of muslin. An aliquot 1 l of filtered rumen fluid was mixed with 4 l McDougall's buffer [31] under anaerobic conditions and maintained at 39 °C using a water bath. About 50 mL of this rumen inoculum was added into the incubation bottles containing the maize and rice straw silages. The incubation bottles were sealed with butyl rubber stoppers and incubated at 39 °C for three incubation periods (0, 6, and 24 h) in triplicate. At each interval, samples were removed from the incubator for the determination of DM digestibility, total gas production, pH, and ammonia-N concentration. The rumen fluid for the in vitro ruminal fermentation characteristics was provided by the Department of Animal Resources, Daegu University, Korea. The management and care of animals were reviewed and approved by the Daegu University Institutional Animal Care and Use Committee (No. 2014-6).

*2.6. Statistical Analysis*

Data were analyzed using General Linear Models (SPSS version 25, IBM, Madison, NY, USA). The effects of lactic acid bacteria (LAB) on maize and rice straw silages were examined using a one-way analysis of variance, and the means were further compared by the Duncan's multiple range test. Principle component analysis (XLstat, Addinsoft, Long Island, NY, USA) was performed to visualize the effects of LAB and fermentation product variations of the maize and rice straw silages using biplots.

**3. Results and Discussions**

*3.1. Effect of Lactic Acid Bacteria on Maize Silage*

The chemical composition of maize silage after the different ensiling periods is presented in Table 1. After the three days ensiling period, there was no significant difference in the chemical compositions among treatments. However, at 40 days, the LAB treatments showed a significant difference ($p < 0.05$)

in DM and OM. The highest DM contents were for the PC, LPL and LA treatments, 32.61%, 33.51% and 33.42%, respectively. The low DM content of the LPA treatment may be a consequence of the fermentation process, which causes DM losses whose magnitude is dependent on the inoculation level and the ensiling period [32].

**Table 1.** Effect of different lactic acid bacteria on the chemical composition (%) of maize silage (dry matter basis unless otherwise stated).

| Days of Ensiling | Treatment (*n* = 3) | DM | OM | CP | NDF | ADF |
|---|---|---|---|---|---|---|
| | NC [1] | 31.58 | 93.28 | 8.71 | 58.24 | 35.95 |
| | PC | 32.54 | 94.19 | 8.45 | 53.22 | 31.26 |
| | LPL | 31.74 | 93.48 | 8.58 | 59.67 | 33.17 |
| 3 | LPA | 31.64 | 93.55 | 8.92 | 54.51 | 34.24 |
| | LA | 31.76 | 93.47 | 8.74 | 55.35 | 36.53 |
| | SEM [2] | 0.353 | 0.366 | 0.240 | 1.919 | 1.630 |
| | *p* | 0.052 | 0.503 | 0.715 | 0.180 | 0.235 |
| | NC | 31.01 | 93.35 | 8.25 | 58.18 | 36.76 |
| | PC | 31.40 | 93.99 | 8.23 | 54.43 | 32.45 |
| | LPL | 31.92 | 93.27 | 8.24 | 57.31 | 36.32 |
| 7 | LPA | 32.01 | 93.67 | 8.33 | 56.86 | 35.68 |
| | LA | 34.59 | 94.68 | 8.54 | 51.97 | 30.05 |
| | SEM | 0.292 | 0.172 | 0.119 | 0.972 | 0.858 |
| | *p* | 0.103 | 0.140 | 0.913 | 0.317 | 0.133 |
| | NC | 30.77 | 92.67 | 8.63 | 59.14 | 37.82 |
| | PC | 27.72 | 93.25 | 8.68 | 54.76 | 33.13 |
| | LPL | 30.61 | 93.98 | 8.72 | 52.71 | 31.73 |
| 20 | LPA | 30.27 | 93.35 | 8.75 | 56.73 | 35.69 |
| | LA | 28.89 | 93.27 | 8.58 | 57.27 | 35.54 |
| | SEM | 0.480 | 0.338 | 0.116 | 1.657 | 1.404 |
| | *p* | 0.585 | 0.191 | 0.844 | 0.141 | 0.080 |
| | NC | 31.28 [a,3] | 93.39 [a] | 8.42 | 58.19 | 36.29 |
| | PC | 32.61 [a,b] | 93.99 [a,b] | 8.28 | 52.90 | 32.15 |
| | LPL | 33.51 [b] | 94.44 [b] | 8.80 | 54.13 | 33.35 |
| 40 | LPA | 31.09 [a] | 93.19 [a] | 8.48 | 57.15 | 37.06 |
| | LA | 33.42 [b] | 94.57 [b] | 8.40 | 50.68 | 30.94 |
| | SEM | 0.198 | 0.117 | 0.051 | 0.823 | 0.686 |
| | *p* | 0.041 | 0.013 | 0.081 | 0.085 | 0.075 |

[1] NC: Negative control (maize silage without any additives); PC: Positive control (maize silage treated with commercial silage additives); LPL: maize silage treated with *L. plantarum;* LPA: maize silage treated with *L. paracasei;* LA: maize silage treated with *L. acidophilus.* [2] SEM: standard error of the mean. [3 a,b,c] Means with different superscripts in the same column are significantly different at $p < 0.05$.

The consumption of carbohydrates by LAB metabolism may be responsible for the DM loss seen in the study [5]. The observed DM loss in this study differed from the observations made by Santos et al. [11] which showed that DM content was not affected by LAB inoculation of maize silage. However, it was found that the utilization of additives was effective in reducing DM losses in only 35% of the studies [11]. DM content was highest in maize subjected to the PC, LPL, and LA treatments. This was expected as *L. plantarum* and *L. acidophilus* are commonly isolated from silage and are also preferred as silage additives [33].

The LPL treatment resulted in a numerically higher CP content in the maize silage compared to the other treatments after 40 days of ensiling ($p = 0.081$). In contrast, several studies have indicated that CP content is not affected by the addition of bacterial inoculant in the silage-making process [11,14,34]. The NDF and ADF contents showed a similar pattern across treatments being low in both the PC and LA treatments. The neutral detergent fiber in the LAB treatments was also not different from the NC treatment. This finding is similar to that reported by Contreras-Govea et al. [9], where no differences

were observed between the control and LAB treated silages. Other studies are in agreement with the present finding, reporting no effect of bacterial inoculants on NDF and ADF [10,34–36]. However, there was an observable decrease in NDF, which can be attributed to the degradation of hemicellulose, which is relatively abundant in maize silage, under the acidic conditions of the ensiling process [37,38].

The pH of silage extracts at the end of the ensiling (40 days) was not affected by the inoculants used (Table 2). Keles and Demirci [39] and Kung Jr et al. [40] concluded that the addition of homofermentative bacteria results in a rapid drop in pH due to the increased production of lactic acid during fermentation. The results of the current study could not confirm this conclusion as all homofermentative bacteria used did not significantly reduce silage pH compared to the NC treatment. On the other hand, several studies seem to concur that bacterial inoculants do have a significant effect on the reduction of pH in silage. An accepted explanation as to why the addition of inoculants sometimes does not show an effect on pH reduction is competition with epiphytic LAB [8], and maize silage is known to have a high epiphytic LAB population [33]. During the ensiling process, homofermentative LAB mainly produce lactate as their primary product of sugar fermentation [41]. All treatments had numerically higher ($p = 0.076$) lactic acid production compared with NC. The LPA treatment numerically had the highest lactic acid production. The acetate present in the LPA and LA treated maize silages after 40 days of ensiling was higher compared to the NC treatment. The lactate to acetate ratios for the LAB treatments were 3.4, 3.5, and 3.0 for LPA, LPL, and LA treatment, respectively. These lactate to acetate ratios were higher than those reported by Huisden et al. [42], however similar to those found by Filya and Sucu [10] and Weinberg and Muck [33] in their studies on LAB-treated maize silage. The ammonia-N as a proportion of total-N was significantly lower ($p < 0.05$) in the NC treatment than the LPA and LA treatments and similar to the PC and LPL treatments. Among the LAB treatments, the LPL treatment recorded the lowest ammonia-N (Table 2). In the determination of spoilage of the silage, it was found that fungal spoilage was significantly lower in maize silage treated with LPL ($p < 0.05$) compared with the other LAB treatments but similar to the NC treatment. The number of lactobacilli was lower ($p < 0.05$) in LPA than that of the rest of the treatments.

**Table 2.** Chemical and microbial analyses of maize silage extracts after 40 days of fermentation.

| Item | Treatment ($n = 3$) | | | | | SEM [2] | *p* |
|---|---|---|---|---|---|---|---|
| | NC [1] | PC | LPL | LPA | LA | | |
| pH | 3.85 | 3.91 | 3.83 | 3.97 | 4.03 | 0.011 | 0.18 |
| Ammonia-N (g/kg total-N) | 0.59 [a,3] | 1.07 [a,b] | 0.76 [a,b] | 1.19 [b,c] | 1.40 [c] | 0.022 | 0.032 |
| Lactic acid (g/kg DM) | 10.89 | 12.63 | 11.72 | 19.4 | 15.97 | 2.057 | 0.076 |
| Acetic acid (g/kg DM) | 3.06 [a] | 4.65 [a,b] | 3.36 [a] | 5.63 [b] | 5.34 [b] | 0.562 | 0.029 |
| Fungi (log CFU/g) | 3.43 [a,b] | 3.61 [b,c] | 3.10 [a] | 3.95 [c] | 3.57 [b,c] | 0.055 | 0.008 |
| Yeast (log CFU/g) | 4.25 [b] | 4.22 [b] | 3.20 [a] | 4.31 [b] | 3.42 [a] | 0.103 | 0.032 |
| Bacillus (log CFU/g) | 6.38 [a,b] | 6.67 [b] | 6.12 [a,b] | 6.00 [a] | 5.81 [a] | 0.078 | 0.044 |
| Lactobacillus (log CFU/g) | 7.61 [b] | 7.56 [a,b] | 7.66 [b] | 7.37 [a] | 7.72 [b] | 0.03 | 0.031 |

[1] NC: Negative control (maize silage without any additives); PC: Positive control (maize silage treated with commercial silage additives); LPL: maize silage treated with *L. plantarum;* LPA: maize silage treated with *L. paracasei;* LA: maize silage treated with *L. acidophilus.* [2] SEM: standard error of the mean. [3 a,b,c] Means with different superscripts in the same column are significantly different at $p < 0.05$.

The effects of LAB inoculated maize silage on in vitro rumen fermentation are shown in Table 3. There was no difference among treatments in dry matter digestibility (DMD). Similar observations were made by Muck et al. [43], where they did not report any differences between LAB-treated silage and the control on DMD after a 24 h in vitro rumen fermentation. Except for the LPA, LAB-treated silages showed numerically higher DMD compared with the NC treatment, although it was not statistically different. A Weinberg et al. [44] study showed that LAB inoculants could potentially improve DMD, which would, in turn, enhance animal performance. Gas production was significantly different ($p < 0.05$) across the treatments after a 24 h incubation period. The NC, LPL and LA treatments

had higher gas production compared to the PC and LPA treatments whose gas production was similar. However, the values of PC and LPA seemed to erroneous relative to their DMDs and it was unclear why such values were obtained. After 6 h of incubation, the pH of treated maize was different ($p < 0.05$) across the treatment. The pH of the PC and LPA treatments was higher ($p < 0.05$) than the rest of the treatments. In the present study, a comparison of the fermentation characteristics during ensiling and in vitro rumen fermentation showed that all LAB used were marginally effective in producing quality maize silages. With lower DM loss and pH, and the lower counts of fungi and yeast, *L. plantarum* appeared to be better for maize silage. These findings are in agreement with previous studies [8,11].

**Table 3.** Effects of LAB treated maize silage on in vitro rumen fermentation.

| Item | Treatment (*n* = 3) | | | | | SEM [2] | *p* |
|---|---|---|---|---|---|---|---|
| | NC [1] | PC | LPL | LPA | LA | | |
| DMD, % | | | | | | | |
| 6 h | 42.63 | 45.71 | 42.39 | 40.17 | 41.09 | 0.711 | 0.216 |
| 24 h | 58.57 | 59.30 | 60.78 | 58.18 | 65.74 | 0.889 | 0.118 |
| pH | | | | | | | |
| 6 h | 6.93 [a,b,3] | 7.00 [c] | 6.97 [b,c] | 6.99 [c] | 6.90 [a] | 0.007 | 0.009 |
| 24 h | 6.79 | 6.78 | 6.79 | 6.78 | 6.75 | 0.007 | 0.265 |
| Total gas production, mL | | | | | | | |
| 6 h | 15.83 | 15.94 | 14.17 | 13.61 | 17.78 | 0.919 | 0.645 |
| 24 h | 52.61 [b] | 38.36 [a] | 58.33 [b] | 29.72 [a] | 57.39 [b] | 1.826 | 0.002 |
| Ammonia-N, mg/100 mL | | | | | | | |
| 6 h | 8.33 [b,c] | 8.02 [a,b] | 8.05 [a,b] | 8.97 [c] | 7.54 [a] | 0.093 | 0.008 |
| 24 h | 7.55 [a] | 8.25 [a] | 8.02 [a] | 9.88 [b] | 8.17 [a] | 0.151 | 0.006 |

[1] NC: Negative control (maize silage without any additives); PC: Positive control (maize silage treated with commercial silage additives); LPL: maize silage treated with *L. plantarum;* LPA: maize silage treated with *L. paracasei;* LA: maize silage treated with *L. acidophilus.* [2] SEM: standard error of the mean. [3 a,b,c] Means with different superscripts in the same row are significantly different at $p < 0.05$.

## 3.2. Effect of Lactic Acid Bacteria on Rice Straw Silage

Rice straw is an agricultural by-product with low nutritive value; it is low in CP but high in lignin and silica contents [45]. Due to its low nutritive value, it fails to meet energy and protein requirements of ruminants [46]. The addition of LAB in rice straw silage has been thought to improve the nutritional value of the rice straw. The chemical composition of rice straw after ensiling for 3, 7, 20, and 40 days is presented in Table 4. There was a significant difference ($p < 0.05$) in DM across the treatments after days 3, 20 and 40. At the end of the 40-day ensiling period, the LPA treatment had the lowest DM compared to the other treatments. The DM contents reported in the current study were similar in range to those reported by Li et al. [47], which were between 28.9 to 39.3%. CP content after 40 days of ensiling was higher ($p < 0.05$) in the LAB treated silages compared to NC treatment. Similar observations were made by Li et al. [47]. This was contrary to the findings of Gao et al. [48], who showed no effects of silage inoculation on CP content. The LA treatment had the least NDF content after 40 days, which was significantly lower than the other treatments except for the PC treatment, which had similar NDF content. This was in line with what was reported by Li et al. [47]. With regards to the difference in NDF content that exists between inoculated and uninoculated silage substrates. Neutral detergent fiber and ADF appear to have been degraded toward the end of the ensiling period. As was seen with NDF, the LA treatment had significantly lower ADF than the rest of the treatments ($p < 0.05$).

**Table 4.** Effect of different starter cultures on chemical composition (%) of rice straw silage (dry matter basis unless otherwise stated).

| Days of Ensiling | Treatments (*n* = 3) | DM | OM | CP | NDF | ADF |
|---|---|---|---|---|---|---|
| 3 | NC [1] | 38.76 [a,b,3] | 90.75 [c] | 3.82 [a] | 75.63 [b,c] | 44.81 [c] |
| | PC | 42.90 [b] | 90.16 [a] | 4.19 [b] | 73.41 [a,b] | 42.90 [b] |
| | LPL | 43.76 [b,c] | 90.47 [b] | 4.28 [b] | 74.92 [b,c] | 43.76 [b,c] |
| | LPA | 45.54 [c] | 90.66 [c] | 4.32 [b] | 77.07 [c] | 45.54 [c] |
| | LA | 40.98 [a] | 90.39 [b] | 4.16 [a] | 71.22 [a] | 40.98 [a] |
| | SEM [2] | 0.251 | 0.026 | 0.040 | 0.413 | 0.251 |
| | *p* | 0.002 | 0.000 | 0.020 | 0.011 | 0.002 |
| 7 | NC | 39.37 | 90.36 [b] | 4.07 | 78.24 | 46.48 |
| | PC | 39.82 | 90.04 [a] | 4.10 | 74.93 | 45.11 |
| | LPL | 41.61 | 90.38 [b] | 4.12 | 74.53 | 44.39 |
| | LPA | 39.80 | 90.40 [b] | 3.95 | 75.08 | 44.77 |
| | LA | 37.66 | 90.01 [a] | 4.12 | 75.90 | 45.06 |
| | SEM | 0.383 | 0.027 | 0.042 | 0.405 | 0.246 |
| | *p* | 0.092 | 0.002 | 0.669 | 0.094 | 0.160 |
| 20 | NC | 36.27 [a,b] | 90.48 [c] | 3.79 [a] | 77.36 [d] | 46.66 [c] |
| | PC | 38.82 [d] | 89.54 [a,b] | 4.24 [b] | 74.32 [c] | 43.52 [b] |
| | LPL | 37.81 [c,d] | 90.21 [b,c] | 4.06 [b] | 71.36 [b] | 42.58 [b] |
| | LPA | 35.82 [a] | 90.26 [b,c] | 4.11 [b] | 76.10 [c,d] | 45.31 [c] |
| | LA | 37.38 [b,c] | 89.46 [a] | 4.08 [b] | 68.46 [a] | 40.20 [a] |
| | SEM | 0.195 | 0.098 | 0.034 | 0.280 | 0.238 |
| | *p* | 0.004 | 0.026 | 0.021 | 0.000 | 0.000 |
| 40 | NC | 36.91 [b] | 90.25 [c] | 3.93 [a] | 73.99 [c] | 44.62 [c] |
| | PC | 38.89 [c] | 89.67 [a,b] | 4.27 [b,c] | 69.07 [a,b] | 40.18 [b] |
| | LPL | 37.71 [b] | 89.97 [b,c] | 4.10 [a,b] | 70.56 [b,c] | 41.15 [b] |
| | LPA | 35.51 [a] | 90.19 [c] | 4.20 [b,c] | 70.73 [b,c] | 41.66 [b] |
| | LA | 37.60 [b] | 89.42 [a] | 4.34 [c] | 66.33 [a] | 37.98 [a] |
| | SEM | 0.139 | 0.067 | 0.027 | 0.486 | 0.388 |
| | *p* | 0.000 | 0.013 | 0.005 | 0.007 | 0.005 |

[1] NC: Negative control (maize silage without any additives); PC: Positive control (maize silage treated with commercial silage additives); LPL: maize silage treated with *L. plantarum*; LPA: maize silage treated with *L. paracasei*; LA: maize silage treated with *L. acidophilus*. [2] SEM: standard error of the mean. [3 a,b,c,d] Means with different superscripts in the same column are significantly different at *p* < 0.05.

The treatments significantly affected the pH of rice straw silage after 40 days (*p* < 0.05, Table 5.). Among the LAB treatments, the PC treatment resulted in the lowest pH, followed by LA and LPA treatments. The NC treatment resulted in a higher (*p* < 0.05) ammonia-N production than the other treatments, though this was similar to the LPL and LPA treatments. Fungi, yeast, and Lactobacillus as determinants of silage spoilage were not statistically significant. The proliferation of the genus Bacillus was significantly affected by the inoculation of the silage substrate (*p* < 0.05), with the LPL treatment having the least effect on Bacillus and the LA treatment having the most. The fermentation characteristics seen in the current study indicate that, except for the LPL treatment, rice straw treated by LAB inoculation produced silage that had lower pH compared to that which was not treated. The depressed pH in the case of LAB inoculated rice straw silage is associated with the production of lactic acid during fermentation. Several studies have also reported similar differences between LAB treated and untreated rice straw silage [47–49].

**Table 5.** Chemical and microbial analyses of rice straw silage extracts after 40 days of fermentation.

| Items | Treatment (*n* = 3) | | | | | SEM [2] | *p* |
|---|---|---|---|---|---|---|---|
| | NC [1] | PC | LPL | LPA | LA | | |
| pH | 5.37 [d,3] | 4.02 [a] | 5.43 [d] | 5.26 [c] | 4.42 [b] | 0.015 | 0 |
| Ammonia-N (g/kg total-N) | 1.28 [b] | 0.74 [a] | 1.00 [a,b] | 1.08 [a,b] | 0.73 [a] | 0.018 | 0.027 |
| Lactic acid (g/kg DM) | 0.97 [a] | 6.01 [c] | 0.83 [a] | 1.04 [a] | 3.38 [b] | 0.69 | 0.001 |
| Acetic acid (g/kg DM) | 2.26 [c] | 0.88 [a] | 1.57 [b] | 2.30 [c] | 1.3 [a,b] | 0.164 | 0 |
| Fungi (log CFU/g) | 3.42 | 3.98 | 3.41 | 3.25 | 3.71 | 0.112 | 0.312 |
| Yeast (log CFU/g) | 5.81 | 5.2 | 4.64 | 5.04 | 4.87 | 0.167 | 0.297 |
| Bacillus (log CFU/g) | 6.49 [b,c] | 6.31 [b,c] | 5.38 [a] | 5.83 [a,b] | 6.79 [c] | 0.09 | 0.004 |
| Lactobacillus (log CFU/g) | 7.55 | 7.38 | 7.67 | 7.56 | 7.7 | 0.052 | 0.361 |

[1] NC: Negative control (maize silage without any additives); PC: Positive control (maize silage treated with commercial silage additives); LPL: maize silage treated with *L. plantarum;* LPA: maize silage treated with *L. paracasei;* LA: maize silage treated with *L. acidophilus.* [2] SEM: standard error of the mean. [3 a,b,c,d] Means with different superscripts in the same row are significantly different at *p* < 0.05.

The in vitro rumen fermentation characteristics of ensiled rice straw are shown in Table 6. The DMD of rice straw silage was different (*p* < 0.05) at 6 and 24 h of incubation due to the treatments. At the end of the 24 h incubation, the PC and LA treatments had the highest DMD compared to other treatments. Our result was in contrast with the study conducted by Chen et al. [50], which had shown no differences in DMD of rice straw treated by *L. acidophilus.* The pH of the ensiled rice straw was significantly affected (*p* < 0.05) by treatment, with the pH of all LAB treatments being lower than that of NC. However, by 24 h of digestion, all treatments had similar pH. There was no significant difference in gas production across all treatments (*p* = 0.157). Ammonia-N differed significantly across the treatments at 6 h of incubation (*p* < 0.05). The LA and LPL treatments had higher concentration of ammonia-N compared to the rest of the treatments. The rice straw silage treated with the LA had the highest CP, and consequently, the amount of protein and amino acids available for deamination onto ammonia-N is likely to be increased. However, another factor that influences the degradation of protein is energy availability [51]. Rice straw has low amounts of energy due to the high level of cell wall content, with some indigestible components (i.e., lignin and silica). Such discrepancy in terms of energy and protein availability to ruminal microorganisms may lead to waste on ammonia-N.

**Table 6.** Effects of LAB treated rice straw silage on in vitro rumen fermentation.

| | Treatment (*n* = 3) | | | | | SEM [2] | *p* |
|---|---|---|---|---|---|---|---|
| | NC [1] | PC | LPL | LPA | LA | | |
| DMD, % | | | | | | | |
| 6 h | 15.69 [a,3] | 20.30 [b] | 14.64 [a] | 15.21 [a] | 18.43 [b] | 0.149 | 0.002 |
| 24 h | 29.71 [a] | 34.40 [b] | 29.00 [a] | 30.38 [a] | 34.36 [b] | 0.092 | 0.012 |
| pH | | | | | | | |
| 6 h | 6.99 [b] | 6.96 [a] | 6.96 [a] | 6.96 [a] | 6.94 [a] | 0.003 | 0.04 |
| 24 h | 6.92 | 6.93 | 6.94 | 6.91 | 6.91 | 0.002 | 0.36 |
| Total gas production, mL | | | | | | | |
| 6 h | 7.63 | 7.67 | 7.57 | 7.4 | 7.13 | 0.119 | 0.964 |
| 24 h | 26.83 | 30.67 | 27.83 | 28.27 | 26.5 | 0.081 | 0.157 |
| Ammonia-N, mg/100 mL | | | | | | | |
| 6 h | 9.09 [a] | 10.74 [b] | 16.80 [d] | 12.46 [c] | 14.80 [d] | 0.238 | 0 |
| 24 h | 9.85 [b,c] | 8.72 [a] | 9.75 [b,c] | 9.01 [a,b] | 10.12 [c] | 0.151 | 0 |

[1] NC: Negative control (maize silage without any additives); PC: Positive control (maize silage treated with commercial silage additives); LPL: maize silage treated with *L. plantarum;* LPA: maize silage treated with *L. paracasei;* LA: maize silage treated with *L. acidophilus.* [2] SEM: standard error of the mean. [3 a,b,c,d] Means with different superscripts in the same row are significantly different at *p* < 0.05.

This study investigated cultures of LAB as inoculums for both maize and rice straw silage. These two silage materials are quite different in chemical composition and other characteristics. Generally, maize silage has relatively lower DM content and higher OM compared to rice straw silage. CP content was higher in maize silage compared to rice straw silage. On the other hand, rice straw silage had higher NDF and ADF values. Both silages showed declining pH after 40 days of fermentation by LAB. With the organic acid compounds, maize silage had higher lactate–acetate ratio than rice straw silage.

### 3.3. Principal Component Analysis

Principle component analysis was conducted to investigate the effects of LAB on the pattern of silage fermentation characteristics. Figure 1 shows the biplots for maize and rice straw silages based on fermentation products (lactic acid, acetic acid, and pH). The biplots divided the observed data into two clusters. The left cluster is mainly comprised of rice straw silage, and the right cluster maize silage. This indicates that the fermentation patterns of rice straw and maize were different. The differences in nutrient content between maize and rice straw silage are probably the reason for these apparent differences. Maize is likely to have a higher starch content compared to rice straw, which is rich in lignin and silica [45,52]. Consequently, maize produced higher amounts of fermentation products compared to rice straw during ensiling, as described in Figure 1.

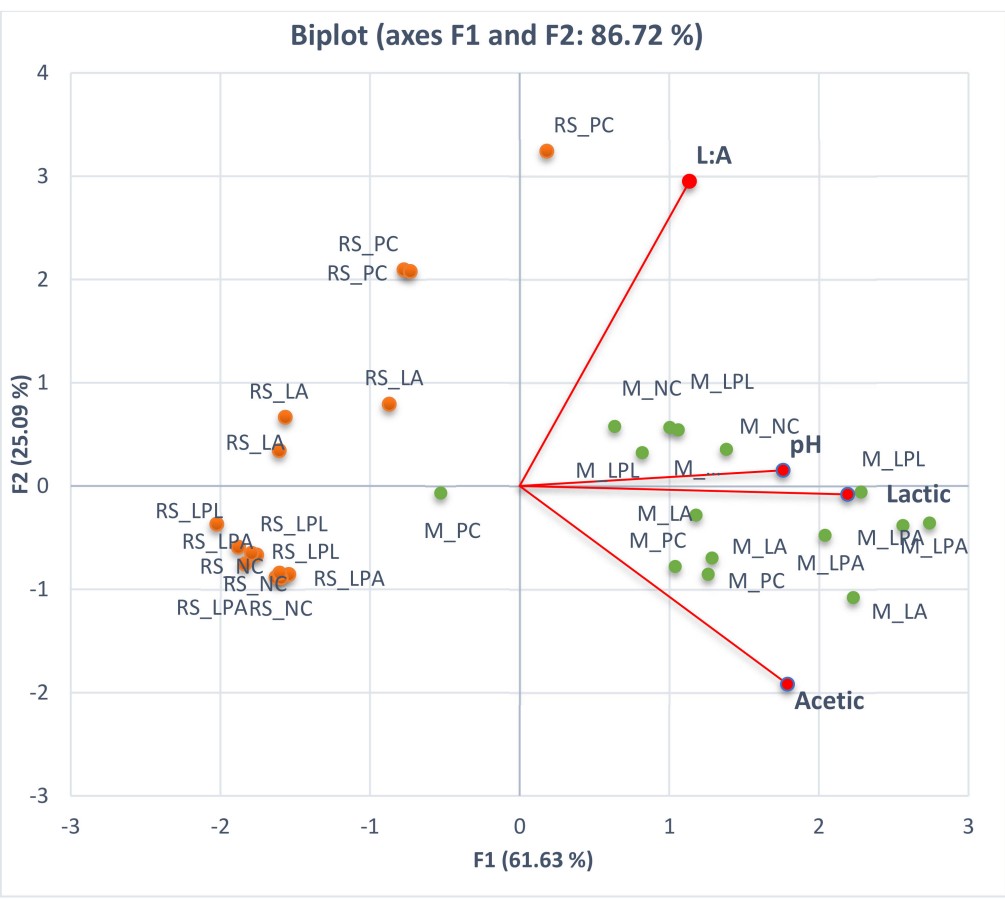

**Figure 1.** Principle component analysis (PCA) of fermentation products (lactic acid, acetic acid, L: A ratio, and pH) in 40 days ensiled maize and rice straw silage. M: maize silage; RS: rice straw, NC: negative control; PC: positive control; LPA: *Lactobacillus paracasei*, LPL: *Lactobacillus plantarum;* LA: *L. acidophilus.*

Such contrast results (see Figure 1) seem to be associated with the nutritive values of maize and rice straw, which will lead to tremendous differences in animal performance. Nevertheless, maize silage is often lacking crude protein, so that either expensive plant protein such as soybean meal or a legume forage in the form of fresh or silage must be supplemented to improve animal performance [52,53]. On the other hand, rice straw as an agricultural byproduct with limited nutritive values plays an important role when forage sources are limited in a country like South Korea [18]. Therefore, improving the quality of either maize silage or rice straw silage, or both is imperative in the future for the ruminant industry.

Another point to highlight was the environmental and genetic variation of forages used. This was well-documented in the study by Jung and Buxtono [54], who reported that the genetic and environmental variation of maize affected the cell wall composition that would be related to the cell wall degradability. Similarly, Bainton et al. [55] reported differences in chemical composition and the degradability of rice straw according to the varieties and environmental conditions. Others [56,57] acknowledged genetic and/or environmental variation of such forages on digestibility in vitro. In this study, only a single cultivar from maize or rice straw was used. However, evidence concerning genetic and environmental factors provides more reason to continue this type of study, examining novel inoculants and their application to diverse varieties even within the same forage species.

## 4. Conclusions

Based on these characteristics of maize and rice straw silage, the LAB, which could effectively ensile each of these substrates, was also different. *L. plantarum* was effective for ensiling maize. *L. plantarum*-treated maize silage had the highest DM and CP contents compared to the other treatments. It also resulted in low ammonia-N concentration compared to the other treatments. On the other hand, for ensiling rice straw, *L. acidophilus* had greater potential as a starter culture. Rice straw silage treated with *L. acidophilus* showed higher DM and CP contents as well as lower NDF and ADF compared to the other treatments. In in vitro rumen fermentation, it also showed a higher DMD compared to the other treatments.

**Author Contributions:** Conceptualization, T.D.M., K.L., J.S., C.H.K., D.Y., S.M.L., J.K., C.L., S.C. and E.J.K.; methodology, T.D.M., K.L., J.S., C.H.K., D.Y., S.M.L., J.K., C.L., S.C., and E.J.K..; software, T.D.M. and E.J.K..; validation, T.D.M. and E.J.K.; formal analysis, T.D.M., K.L., J.S. and E.J.K.; investigation, T.D.M. and E.J.K.; resources, C.H.K., D.Y., S.M.L. and E.K.; data curation, T.D.M. and E.J.K.; writing—original draft preparation, T.D.M. and E.J.K.; writing—review and editing, T.D.M., K.L., J.S., C.H.K., D.Y., S.M.L., J.K., C.L., S.C. and E.J.K.; visualization, T.D.M. and E.J.K.; supervision, E.J.K.; project administration, E.J.K.; funding acquisition, J.K., C.L. and E.J.K. All authors have read and agreed to the published version of the manuscript.

**Funding:** This research was financially supported by the Ministry of Trade, Industry and Energy (MOTIE) and Korea Institute for Advancement of Technology (KIAT) through the Research and Development for Regional Industry (No. R0002505).

**Acknowledgments:** The rumen fluid for the in vitro ruminal fermentation characteristics was provided by the Department of Animal Resources, Daegu University, Korea. The management and care of animals were reviewed and approved by the Daegu University Institutional Animal Care and Use Committee (No. 2014-6). We greatly appreciate the technical support of Daegu University (Chang Weon Choi and his staff).

**Conflicts of Interest:** The authors declare no conflict of interest.

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
