# Peer review of "Effect of Lactic Acid Bacteria on the Nutritive Value and In Vitro Ruminal Digestibility of Maize and Rice Straw Silage"

_applsci, doi:10.3390/app10217801_

Round 1

Reviewer 1 Report

The manuscript entitled „Effect of Lactic Acid Bacteria on the Nutritive Value and In Vitro Ruminal Digestibility of Maize and Rice Straw Silage” by Tabita Dameria Marbun et al. is a well written article with interesting results, but I have some concerns:

  1. The article is based on some quite old experiments - 7 years ago, if I understood correctly: Line 60-61: “Experiment 1 was conducted in Gunwi-gun, Korea, in August 2013 using ripe, yellow maize, while Experiment 2 was carried out  in Cheongni-myeon, Sangju, Korea, in October 2013 using rice straw as the substrate.” I am curious when were the chemical analysis and the other determination  performed. How much time were this samples stored?
  2. The introduction section is quite short. Please consider introducing some information regarding maize and rice straw nutritive value, chemical composition and overall importance.
  3. According to Instructions for Authors the section Results should be treated separately from Discussion (use Microsoft Word template or LaTeX template to prepare your manuscript, found at Applied sciences website).

Author Response

Authors’ response to the comments from the reviewers

We sincerely appreciate the time and efforts that reviewers have dedicated to providing valuable feedback to our manuscript. We have made some revisions according to the suggestions and comments from reviewers. Due to changes made in the original manuscript, the line numbers are likely to differ from the original manuscript. For convenience, the changes in the manuscript are made via the ‘track change’ of Word.

Reviewer 1:

The manuscript entitled „Effect of Lactic Acid Bacteria on the Nutritive Value and In Vitro Ruminal Digestibility of Maize and Rice Straw Silage” by Tabita Dameria Marbun et al. is a well written article with interesting results, but I have some concerns:

  1. The article is based on some quite old experiments - 7 years ago, if I understood correctly: Line 60-61: “Experiment 1 was conducted in Gunwi-gun, Korea, in August 2013 using ripe, yellow maize, while Experiment 2 was carried out in Cheongni-myeon, Sangju, Korea, in October 2013 using rice straw as the substrate.” I am curious when were the chemical analysis and the other determination performed. How much time were this samples stored?

(Authors’ response) We understand the reviewer’s concern. All the experimental progress and the chemical analyses including in vitro rumen simulation study were completed in 2014 immediately after the silage preparation. Therefore, the silage samples were only stored until further analysis (less than a few months).

Due to various activities that I have been involved in as a newly-employed assistant professor, I (as a corresponding author) was not able to develop the results of the current study into a manuscript up until recently. Submitting to the special edition of the journal ‘Applied Sciences’ provides us an excellent opportunity to publish our work.

  1. The introduction section is quite short. Please consider introducing some information regarding maize and rice straw nutritive value, chemical composition and overall importance.

(Authors’ response) We agree with the reviewer. According to the reviewer's recommendation, additional information on maize and rice straw were provided in the Introduction section (Please see p. 2, lines 51-52 and 54-58).

  1. According to Instructions for Authors the section Results should be treated separately from Discussion (use Microsoft Word template or LaTeX template to prepare your manuscript, found at Applied sciences website).

(Authors’ response) Thank the reviewer for this suggestion. We have carefully examined the instructions for authors of the MDPI journal.

(https://www.mdpi.com/journal/applsci/instructions)

It says that the “Discussion section may be combined with Results.”

We thought it might be more logical to combine the Results and Discussion with two contract substrates, delivering our findings to the reader. Therefore, we decided to put the Discussion section together with Results to understand the results of this study better.

Reviewer 2 Report

The effects of bacterial inoculants on Maize has been published in hundreds of articles.  You cite only a very few.  when discussion your results you should compare to many (perhaps % of total research reports).

There is really nothing new in the study with inoculants on corn silage.  I suggest that you focus on the straw data (which is worthwhile) and either not mention the maize data or mention briefly only to use as a check.

The materials and methods needs to be improved. Was only one sampling of the rice staw taken? if so, what about an genetic and environmental variation?

You mention frozen samples but there is no description of how the samples were frozen and storage temperature.

You mention rumen fluid from two animals. What was the diet of the animals.  Was the rumen fluid mixed to run the in vitros (why not run separately and have animal variation included in model).  We usually recommend at least 3 animals.

Statement in line 132 is incorrect.  Acids produced in fermentation are still considered part of the dry matter.

Author Response

Authors’ response to the comments from the reviewers

We sincerely appreciate the time and efforts that reviewers have dedicated to providing valuable feedback to our manuscript. We have made some revisions according to the suggestions and comments from reviewers. Due to changes made in the original manuscript, the line numbers are likely to differ from the original manuscript. For convenience, the changes in the manuscript are made via the ‘track change’ of Word.

Reviewer 2:

The effects of bacterial inoculants on Maize has been published in hundreds of articles. You cite only a very few. When discussion your results you should compare to many (perhaps % of total research reports).

(Authors’ response) We understand the reviewer’s comment and thank the reviewer for the suggestion. For the maize silage discussion and their inoculants applied, some 14 references were cited in the original manuscript. Following the reviewer’s recommendation, another five references are cited below:

  1. O'Mara, F. P.; Fitzgerald, J. J.; Murphy, J. J.; Rath, M., The effect on milk production of replacing grass silage with maize silage in the diet of dairy cows. Livest. Prod. Sci. 1998, 55 (1), 79-87. [reference number 15]

  1. Khan, N. A.; Yu, P.; Ali, M.; Cone, J. W.; Hendriks, W. H., Nutritive value of maize silage in relation to dairy cow performance and milk quality. J. Sci. Food Agric. 2015, 95 (2), 238-252. [reference number 16]

  1. Shaver, R.; Erdman, R.; O’connor, A.; Vandersall, J., Effects of silage pH on voluntary intake of corn silage and alfalfa haylage. J. Dairy Sci. 1985, 68 (2), 338-346. [reference number 17]

  1. Bernardi, A.; Härter, C. J.; Silva, A. W. L.; Reis, R. A.; Rabelo, C. H. S., A meta‐analysis examining lactic acid bacteria inoculants for maize silage: Effects on fermentation, aerobic stability, nutritive value and livestock production. Grass Forage Sci. 2019, 74 (4), 596-612. [reference number 34]

  1. Kung Jr, L.; Stokes, M. R.; Lin, C., Silage additives. In Silage Science and Technology, Buxton, D. R.; Muck, R. E.; Harrison, J. H., Eds. American Society of Agronomy: Madison, Wisconsin, USA, 2003; Vol. 42, pp 305-360. [reference number 40]

There is really nothing new in the study with inoculants on corn silage. I suggest that you focus on the straw data (which is worthwhile) and either not mention the maize data or mention briefly only to use as a check.

(Authors’ response) Thank the reviewer for the comments. We agree that there is a large body of literature existed for maize silage over decades. One of the purposes of this study was to examine how different LAB strains inoculated during the ensiling process affect the nutritive value and in vitro digestibility of maize and rice straw silages, as indicated in p. 2, line 58-61. This study showed that two contrast substrates with different nutritive values had different fermentation patterns depending on the LAB added before the ensiling period. Therefore, maize silage-related information may be necessary in order to explain clearly the results of this study.

The materials and methods needs to be improved. Was only one sampling of the rice staw taken? if so, what about an genetic and environmental variation?

(Authors’ response) We understand that plants harvested at different times of year give you different nutritive values, and therefore, rice straw harvested as silage is likely to be affected by the environment at the time. Such variation is also occurred by genetic differences.

This is why this type of experiment needs to be repeated with different varieties or cultivars, even within the same forage type, to provide more accurate information on silage inoculants. In our study, the maize (Zea mays L.) cultivar was ‘Kwangpyeongok,’ and rice straw was from Oryza sativa subsp. Japonica, cultivar Ilpum. This information was additionally described in the text (please see lines of 65-68).

We wish to conduct more studies in the near future to produce better quality silages with maize and rice straw, and also newer or different inoculants are hoped to be examined as well. Thank the reviewer again.

You mention frozen samples but there is no description of how the samples were frozen and storage temperature.

(Authors’ response) The information related to how the sample was frozen and storage temperature was described in p. 2, line 84.

You mention rumen fluid from two animals. What was the diet of the animals. Was the rumen fluid mixed to run the in vitros (why not run separately and have animal variation included in model). We usually recommend at least 3 animals.

(Authors’ response) For the rumen fluid collection to conducted an in vitro rumen simulation study, two cannulated animals were used, and their diet information is described on page 3, lines 116-118.

Indeed, we pooled rumen fluid from two animals, and this is typified in the area of in vitro rumen simulation study to get representative rumen microflora. This type of in vitro rumen fermentation study is quite common in feed evaluation for ruminants. For example, Yanez-Ruiz et al. (2016. Design, implementation and interpretation of in vitro batch culture experiments to assess enteric methane mitigation in ruminants—a review. Animal Feed Science and Technology, 216, 1-18) published an excellent review on in vitro batch culture, and they recommended “using 3 or more (optimum) or 2 (minimum) animals as donors of rumen inoculum.” There is more evidence in the literature that at least 2 animals should be used to get representative rumen fluid (i.e., Cone et al., 1999. Journal of Dairy Science 82:957-966).

In terms of separating rumen fluid, i.e., using individual rumen fluid is a different issue. In this matter, an “animal variation” should be included as a factor and statistically examined which is somewhat different from the purpose of this study. Further concerns may be discussed, and it is again notable from Yanez-Ruiz et al. (2016) that

“Muetzel et al. (2014) compared rumen fluid from cattle (Holstein × Jersey cows) and sheep using a newly developed automated in vitro system and reported that total gas production is unaffected by donor animal species.”

Statement in line 132 is incorrect. Acids produced in fermentation are still considered part of the dry matter.

(Authors’ response) Thank you for your comment. The statement in line 132 is incorrect and unnecessary. Thus, we decided to delete that line.

Reviewer 3 Report

The manuscript of  Tabita Dameria Marbun et al. titled “ Effect of Lactic Acid Bacteria on the Nutritive Value and In Vitro Ruminal Digestibility of Maize and Rice Straw Silage” is well written but needs some minor changes and improvements.

  1. All Latin names of bacteria should be given in full upon first use, and then abbreviated throughout the manuscript.
  2. Is silage were sterile before adding bacteria cultures? I assumed not, so in Tables 3 and 5 the authors count total lactobacillus CFU. This information should be added, and in tables write Lactobacillus sp. Bacillus sp. ….What species of fungi were determined? Similar for yeast.
  3. In all tables please add information about samples number used for analysis
  4. Line 87, add information about pH apparatus – model, type of electrode…
  5. Line 77, add condition used to ground the samples – temp, time…..
  6. Line 105 “In vitro rumen fermentation was conducted using the Tilley and Terry”, You used methods described by Tilley and Terry, not Tilley and Terry – please correct. So please carefully read the manuscript and correct English punctuations, article usage, and this kind of mistake as has been described above.
  7. Figure 1, What is the n value? It should be clearly described in the figure description, moreover, the figure is not clear, maybe you will use a different color for species and silage and add color legend or description without letters
  8. The authors should discuss their results concerning the animal food requirements

The authors analyzed many parameters at different times, so my question is they analyzed lactic and acetic acid content after 7 and 20 days if not why? if yes, please discuss.

Author Response

Authors’ response to the comments from the reviewers

We sincerely appreciate the time and efforts that reviewers have dedicated to providing valuable feedback to our manuscript. We have made some revisions according to the suggestions and comments from reviewers. Due to changes made in the original manuscript, the line numbers are likely to differ from the original manuscript. For convenience, the changes in the manuscript are made via the ‘track change’ of Word.

Reviewer 3

The manuscript of Tabita Dameria Marbun et al. titled “ Effect of Lactic Acid Bacteria on the Nutritive Value and In Vitro Ruminal Digestibility of Maize and Rice Straw Silage” is well written but needs some minor changes and improvements.

  1. All Latin names of bacteria should be given in full upon first use, and then abbreviated throughout the manuscript.

(Authors’ response) We agree with the reviewer. We have changed the Latin names of bacteria to be abbreviated (for detail: p. 1, line 18-19, p. 2, lines 73-74, p. 10, line 312).

  1. Is silage were sterile before adding bacteria cultures? I assumed not, so in Tables 3 and 5 the authors count total lactobacillus CFU. This information should be added, and in tables write Lactobacillus sp. Bacillus sp. ….What species of fungi were determined? Similar for yeast.

(Authors’ response) Thank you for your valuable and thorough comment. It would have been interesting to show the original (epiphytic) bacteria contained in the substrates. Unfortunately, we did not have the data related to the epiphytic bacteria of substrates. The species of fungi and yeast were unknown as we cultivated fungi and yeast using general media, which provide total counts only.

  1. In all tables please add information about samples number used for analysis

(Authors’ response) Thank you for your recommendation. The information related to the sampling number was already described in the text on the Materials and Methods section (please see p. 2, line 76). Additionally, sample numbers are indicated in all tables.

  1. Line 87, add information about pH apparatus – model, type of electrode…

(Authors’ response) We already provided the probe model's information on page 3, line 94-96.

  1. Line 77, add condition used to ground the samples – temp, time…..

(Authors’ response) Thank the reviewer for the comment. Unfortunately, we do not have the data related to temperature and time related to the grinding condition. Instead, the samples were ground to be small enough to pass a 1-mm screen using ultra-centrifugal mill ZM 2000 (Retch, Haan, Germany). We revised the text with more accurate information (Please see page 2, line 83).

  1. Line 105 “In vitro rumen fermentation was conducted using the Tilley and Terry”, You used methods described by Tilley and Terry, not Tilley and Terry – please correct. So please carefully read the manuscript and correct English punctuations, article usage, and this kind of mistake as has been described above.

(Authors’ response) Thank the reviewer for the comment. We agree with the reviewer and have revised line 105 (after revision, line 114).

  1. Figure 1, What is the n value? It should be clearly described in the figure description, moreover, the figure is not clear, maybe you will use a different color for species and silage and add color legend or description without letters

(Authors’ response) We agree with the reviewer’s comment. We have modified the Figure by changing the legend's color to be easily understood (Please see the revised Figure on page 10).

  1. The authors should discuss their results concerning the animal food requirements

(Authors’ response) Thank the reviewer for the comment.

In the end, silages of maize or rice straw will be used as animal diets, and therefore it may be appropriate to discuss the results of the current study concerning animal feed requirements. Without proper in vivo studies, it may be somewhat challenging to deepen our discussion. Accepting the reviewer's recommendation, we have supplemented the text below towards the end of our paper (please see page 9, line 301-308).

“Such contrast results (see Figure 1) seem to be associated with the nutritive values of maize and rice straw, which will lead to tremendous differences in animal performance. Nevertheless, maize silage is often lacking crude protein, so that either expensive plant protein such as soybean meal or a legume forage in the form of either fresh or silage must be supplemented to improve animal performance [52,53]. On the other hand, rice straw as an agricultural byproduct with limited nutritive values plays an important role when forage sources are limited in a country like South Korea [18]. Therefore, improving the quality of either maize silage or rice straw silage, or both is imperative in the future for the ruminant industry.”

  1. The authors analyzed many parameters at different times, so my question is they analyzed lactic and acetic acid content after 7 and 20 days if not why? if yes, please discuss.

(Authors’ response) Thank the reviewer for the comment. Indeed, the study prepared samples at days 7 and 20 from the moment of silage preparation, and yet those samples were not adequately analyzed in terms of silage fermentation characteristics (i.e., pH, lactate, acetate, etc.). With limited time and resources, we were not able to conduct chemical analyses on those time points. Instead, the proximate analysis was performed, which was less labor-intensive. It would have been very interesting to analyze samples from 7- and 20-day ensiling and compare them with samples from 40-day ensiling. We did not revise the main text for this comment. 

Round 2

Reviewer 2 Report

I still did not see the frozen storage temperature of material

When I asked for description of the inoculum donor diet, I meant to list the species of forage in the diet since the inoculum microbial composition is determined by the donor diet forage.

I am still concerned that little recognization is given to environmental and genetic variation for results.
